# Promoting the Social Inclusion of Children with ASD: A Family-Centred Intervention

**DOI:** 10.3390/brainsci10050318

**Published:** 2020-05-25

**Authors:** Roy McConkey, Marie-Therese Cassin, Rosie McNaughton

**Affiliations:** 1Institute of Nursing and Health Research, Ulster University, Newtownabbey BT37 0QB, UK; 2Cedar Foundation, Belfast BT6 8RB, UK; mt.cassin@cedar-foundation.org (M.-T.C.); r.mcnaughton@cedar-foundation.org (R.M.)

**Keywords:** ASD, autism, families, social inclusion: home-based, Ireland

## Abstract

The social isolation of children with autism spectrum disorder (ASD) is well documented. Their dearth of friends outside of the family and their lack of engagement in community activities places extra strains on the family. A project in Northern Ireland provided post-diagnostic support to nearly 100 families and children aged from 3 to 11 years. An experienced ASD practitioner visited the child and family at home fortnightly in the late afternoon into the evening over a 12-month period. Most children had difficulty in relating to other children, coping with change, awareness of dangers, and joining in community activities. Likewise, up to two-thirds of parents identified managing the child’s behaviour, having time to spend with other children, and taking the child out of the house as further issues of concern to them. The project worker implemented a family-centred plan that introduced the child to various community activities in line with their learning targets and wishes. Quantitative and qualitative data showed improvements in the children’s social and communication skills, their personal safety, and participation in community activities. Likewise, the practical and emotional support provided to parents boosted their confidence and reduced stress within the family. The opportunities for parents and siblings to join in fun activities with the child with ASD strengthened their relationships. This project underscores the need for, and the success of family-based, post-diagnostic support to address the social isolation of children with ASD and their families.

## 1. Introduction

Internationally, there has been a marked rise in the number of children diagnosed with an autism spectrum disorder (ASD) [1]. Early intervention is an agreed priority to ameliorate the main symptoms as soon as the condition is identified, especially in early childhood [2]. Even so, a national UK survey of more than 1000 parents found that nearly half (46%) received no follow-up appointment after the diagnosis was made and only 21% of parents received a direct offer of help/assistance. Additionally, more than 60% of parents expressed dissatisfaction with post-diagnostic support and only 5% were very satisfied with it [3]. 

A particular concern of parents is the lack of social and communication skills experienced by children with autism [4]. This often results in difficulties in interacting with their peers and isolation from community activities. Interventions aimed at promoting community participation have proved effective and a systematic review identified the factors that contributed to their success. This included facilitating “friendships alongside recreational participation, include typically developing peers, consider the activity preferences of children and adolescents in developing programmes, and accommodate individual impairments and needs through grading and adaptive leisure activities” [5] (p. 825).

Families are the primary caregivers of children with ASD. Tint & Weiss [6] in their scoping review noted that considerable research had detailed the correlates of the chronic physical, emotional, social, and financial stressors these families experience. They concluded that “a better understanding of family well-being of individuals with ASD is essential for effective policy and practice” (p. 262). A review of international research to date has identified promising strategies for supporting families [7] as well as targeting parental challenges such as stress, depression, and self-efficacy. “It may be especially constructive to provide wraparound services for families, in which resources and supports are provided (i.e., parent training, therapeutic services, respite care, social services, family counselling) in addition to developmental and behavioural services for the child” (p. 72). 

Moreover, the impact on non-disabled siblings is worthy of attention given the increasing evidence that they fared worse in terms of psychological functioning, internalized behaviour problems, social functioning, and sibling relationships while also showing increased anxiety and depression [8]. 

A Canadian study into family quality of life (FQOL) who have a child with ASD found that opportunities to engage in leisure and recreation activities for the family as a whole was associated with an increased FQOL [9]. However, many families are not engaging in these activities on a regular basis. These researchers recommended that: “service providers could offer leisure and therapeutic recreation options to families, while they wait for other therapeutic services in order to provide additional options to families with a child with ASD.” (p. 340).

Tint & Weiss [6] also identified the value of parents meeting other parents. Research studies indicate that socially isolated mothers may experience greater stress and have fewer socially satisfying interactions with their children. “Participating in group interventions may be beneficial for parents of children with ASD because it provides them with an opportunity to connect with other parents who are having similar experiences” (p. 94).

To date, most research studies have been undertaken with better educated, more affluent parents and little attention has been paid as to how ‘under-resourced’ families (those with low incomes and limited education) can be better supported when a child has ASD. A small-scale study in the USA found that “specific strategies to increase participant retention and decrease attrition included providing sessions in home, reducing travel requirements... and providing community resource support. One of the strengths was the presence of a strong referral system... and a team committed to helping patients access ASD-specific services” [10] (p. 94). More generally, a systematic review of home visiting programs for disadvantaged families concluded that: “home visitation by paraprofessionals is an intervention that holds promise for socially high-risk families with young children” [11] (p. 1).

The foregoing review provided the rationale for the family-centred, post-diagnostic support service to families and children aged 4 to 11 years with ASD in rural counties of Northern Ireland. The main focus was on promoting the social inclusion of the child and family within their local communities.

## 2. Materials and Methods

The project was conceived and delivered by the Cedar Foundation, which is a voluntary, non-profit organisation with a long history of delivering services to people with disabilities and their families. Charitable funding was obtained from the UK Big Lottery Fund for a five-year period. 

A logic model was developed to guide the design and implementation of the service as well as its evaluation (see Figure 1). The model summarises the theory of change as to how the intervention would produce the intended outcomes in the short term and the possible longer-term consequences for the child and family [12]. 

### 2.1. Description of Inputs and Activities 

Five project workers including one full-time and two part-time job shares each covered one of three counties in the western part of Northern Ireland, which is largely rural with a higher proportion of under-resourced families. All staff had a bachelor’s degree in psychology plus a minimum of one year of paid experience. In addition to their qualifications and experience in autism, they received further training in ASD during the course of the project. Fortuitously, the appointed workers lived in the county in which they worked and were familiar with the community resources available there. 

The project workers were line managed through Cedar’s Community Services Manager who also managed Cedar’s other projects in the western area. Links with these projects provided project workers with further support and training. 

Each family received fortnightly visits for a 12-month period. However, all families were given opportunities to maintain contact with the project and to participate in all future group activities.

Quarterly meetings were held in each county between Cedar staff and the social workers from the children’s ASD multi-disciplinary team of the statutory Health and Social Care Trust who undertake assessment and diagnosis of ASD. Potential referrals of families to the project were discussed with the social workers and the ongoing case load of families and children reviewed. An extension of time on the project was agreed for those families who had continuing needs. Similarly, families deemed to have higher needs were given priority when a place became available on the project. 

A project worker visited the child and family at home in the late afternoon into the evening once every two weeks on average. The first visits were used to assess the child and family needs and agree on an individual plan for meeting those needs. The project worker devised and implemented learning activities to address the children’s needs. These occurred within the family home or on outings to community locations and activities. The aim was to embed the learning in real-life settings, which schools are often unable to do.

Project staff made learning aids, such as visual schedules or story books. These resources were left for the families to use. 

The visits also provided opportunities to advise and guide the families on managing the child’s behaviour as well as furthering their learning. As the relationship with the project worker developed, families became more open about further issues and worries they had. As well as providing emotional support, the workers signposted families to other services in their locality. 

The home visits were at a time when the project workers met other family members such as siblings and fathers. If appropriate, siblings were also invited to join in the activities the project worker undertook with the child, with the goal of enhancing the child’s inclusion in family activities. 

The project worker introduced the child to community activities in line with their learning targets and wishes. These provided opportunities to teach road safety or social interactions with other children as well as introducing the child to leisure activities such as swimming, horse-riding, football, and youth clubs.

Project workers also made contact with schools if required but especially for children with ASD who were soon to transfer from primary to secondary school. This gave opportunities for devising common approaches across home and school settings. These visits have led to increased contact between the schools and families. 

Family Fun days were organized in each county four times a year and families were invited to attend. Siblings were especially welcome along with the fathers and mothers. They were held in community locations such as leisure centres or soft-play facilities with a range of activities organized to provide social interactions among the children and among the parents. The intention was also to introduce families to locations to which they could take their children in the future.

Parent Networking meetings were organized in two counties since there was less interest in a third county where parents already had access to other parent groups. An invited speaker talked on a topic of interest or else ‘pampering sessions’ for mothers took place.

Sibling groups were also provided in one county as a trial that brought together the brothers and sisters for play activities, but they also provided opportunities for them to learn more about autism and how they could respond to their sibling’s behaviours.

### 2.2. The Characteristics of Families and Children Involved with the Project 

In all, 92 families with 96 children with ASD were involved with the project over a four-year period. 

One quarter of families (25%) had a lone parent, which is higher than the Northern Ireland average of 18%. More than two-thirds had a wage earner in the family but more than one-quarter were dependent on social security benefits. This is also higher than the average for Northern Ireland, which is 16.1%. Around half of the families (51%) lived in the top 30% of socially deprived areas in Northern Ireland with very few families coming from more affluent areas. Thus, the project had recruited and retained under-resourced families, which was its intention. 

Of the 96 children, 76 (79.2%) were male and 20 (20.8%) were female. The median age when they joined the project was 7.7 years (range of 3.4 to 11.8 years).

A small proportion of the children were enrolled in a special school or special unit attached to a mainstream school (14.5%) but most attended their local primary school (85.5%). However, 79 children (82.3%) had a statement of special educational needs and others were in the process of being assessed for such. Seven children (7.3%) had an additional learning difficulty. This group were enrolled at the start of the project but, in later years, children with a learning disability and autism were not referred to the project. 

More than one-third of the children (35.5%) were an only child. Three families had two children with ASD who also participated in the project.

Further details of the difficulties experienced by the children with ASD are given below.

### 2.3. Evaluation of Project Outcomes

The first author was the external evaluator and a mix of qualitative and quantitative descriptive data was collected. The qualitative data was obtained through face-to-face interviews conducted with all the project staff and their managers (seven in all) and the three social workers who referred families to the project. Seven parents were interviewed at one of two family events. In addition, telephone interviews were conducted with 10 parents chosen by project staff to represent the range of children and families involved in the project across the three counties. Self-completion questionnaires requesting their views on the project and perceived outcomes were completed by a further 15 families who responded to an invitation sent to all parents by project staff as a text message. 

The quantitative data was obtained through two rating scales that were developed in association with the project staff with one relating to the child and another relating to the family (see Table 1 and Table 2 in the results section). They provided a means for assessing each child’s difficulties at the start of their involvement and the outcomes achieved as a result of their involvement. Similarly, family needs could be identified and outcomes could be assessed. The project staff completed both rating scales based on their records for all the families with whom they had been involved and the reviews they had undertaken with them during and at the end of the home visits. 

Formal ethical approval was not required since this study was deemed an evaluation of an ongoing service and not a research project according to Guidance from the UK Health Research Authority. Families gave consent for the information they provided directly or indirectly to be used anonymously in any reports on the project internally and externally, such as to project funders. 

The evaluation was completed at the end of December 2019. 

Due to time and resource constraints, information was not obtained from the children about the project nor were any external assessments available of their developmental progress. Moreover, there was no comparable group of children and families who did not receive the service since this was not ethically feasible for Cedar Foundation to undertake. 

## 3. Results

### 3.1. Children’s Difficulties and Outcomes

The project staff who had been involved with the children and families helped to devise a summary tool for assessing each child’s difficulties at the start of their involvement and the outcome achieved in relation to them. Table 1 lists the items and the ratings provided by the project worker across the 96 children, but some items were omitted for some individuals due to uncertainty or irrelevance for the child or family. Hence, the totals do not add to 96. The percentages are calculated on the number of ratings made. 

The first column of Table 1 describes the issues that were of concern to families about their child with ASD at the start of their involvement with the project. The lower the percentage, the more children for whom the difficulty was identified. In all, eight of the sixteen listed difficulties were ones affecting the majority of families. 

Difficulties in relating to other children affected all of the children in the project. The next most common cluster of difficulties related to awareness of dangers, difficulty with change, and joining in community activities with over seven in eight children affected by them. A cluster of emotional reactions was the next most common and this included anxiety, extreme fear and nervousness, anger, and meltdowns. In addition, more than 50% of children had problems with following instructions. 

By contrast, some difficulties were identified by fewer than one in five children even though they include behaviours commonly associated with ASD such as an unusual response to something new, unusual interest in toys or objects, problems with play, and keeping themselves occupied. 

Overall, the median number of issues identified as being a difficulty for the child was eight (range of 3 to 14). 

### 3.2. Changes in the Children

Columns 2 and 3 in Table 1 indicate the outcomes from the help provided by the project. On all items, the difference in the ratings was significantly different from what would be expected by chance (Chi Square tests *p* < 0.001). Column 2 indicates difficulties that had improved since participating in the project and which were now considered less of a problem. The majority of children had improved on the six most commonly mentioned difficulties including relating to other children, awareness of dangers, coping with change, joining in community activities, and managing anxieties and fears. In all the median number of difficulties on which children were deemed to have improved was seven (range 1 to 13). 

Column 3 indicates the difficulties that remained a problem even though project staff had addressed it and, as such, they represent a continuing need for children and families. The most common – albeit for only one in five children or less - were the top three items listed in Table 1, which include notable difficulty in relating to other children, in awareness of danger, and in managing change. Overall, most children had no difficulties that were a continuing problem, but others had up to 10 difficulties that continued.

### 3.3. Issues for Families and Outcomes

The issues that commonly face families who have a child with ASD were listed in a similar rating scale to that used for the child’s difficulties. Table 2 lists the items and the ratings provided by the project worker.

The first column of Table 2 indicates the issues that were of concern to families at the start of their involvement with the project. The lower the percentage, the more families for whom the issue was identified. In all, nine of the 14 were issues that affected the majority of families. Having knowledge about the services and supports available was the most common and identified in 96% of families. Up to two-thirds of families identified managing the child’s behaviour, having time to spend with other children, and taking the child out of the house as the main issues. 

By contrast, three issues affected 20% of parents or less including family quality of life, main caregiver being anxious or depressed, and having people to turn to if a problem arose. Overall, families were presented with a median of seven different issues (range from 0 to 11 issues identified). 

### 3.4. Outcomes for Families

Columns 2 and 3 of Table 2 indicate the outcomes from the help provided by the project. On all items, the difference in the ratings was significantly different from what would be expected by chance (Chi Square tests, *p* < 0.001). The second column indicates issues that were deemed to be no longer an issue for families. The two issues that a majority of families (50% and over) benefited from were: knowing the supports and services available and having time to spend with other children. In addition, more than two in five families also gained from communication with schools, the child going out of the house, meeting other parents, and finding activities that the whole family can join in. In all, families had a median of five issues resolved (range from 0 to 11).

### 3.5. Perceptions of Project Outcomes and Impact 

The perceptions of three groups of stakeholders were sought regarding the outcomes of the project for the children and for the families as a whole: namely parents (*n* = 16), project staff (*n* = 7), and social workers who had referred children to the project (*n* = 4). This information was obtained mostly through one-to-one interviews, which were audio-recorded and transcribed and were supplemented by self-completed questionnaires. A thematic content analysis using the framework proposed by Braun & Clarke [13] was undertaken. The initial codes derived from the responses were grouped under two core themes: the impact on the child and the impact on families. Table 3 summarises the subthemes along with supporting quotations from parents that the other respondents confirmed. The person quoted is noted in brackets. 

All the stakeholders recounted various impacts that the project had on the children, which elaborated the changes noted in Table 1. Enhanced social interactions were a common outcome since the children learned to overcome their difficulties in interacting with their siblings and other children. Improvements in the children’s behaviours were also noted with fewer meltdowns and less anger. The acquisition of new skills was confirmed especially those needed when accessing community facilities. 

The stake-holders also confirmed the impact of the project on families. Parents had learned new ways of interacting with their child from the project workers. The changes they saw in their child boosted their confidence and this, coupled with the opportunity to have some free time to spend with their other children, resulted in them feeling less stressed. They also appreciated the opportunity to meet with other parents.

#### 3.5.1. The Most Successful Aspects of the Project

Not surprisingly, the stake-holders identified different aspects of the project that they considered to be particularly successful. To some extent, this reflects the diversity of needs that families and children present and the flexibility of the project in meeting their needs. For most though, the one-to-one work with the children were the most frequently cited.


*She had a great rapport with him, you know, she really met him at his level and he never said I don’t want her coming, never, never did. She has a fantastic rapport with him.*
(Mother JU)

Listening to families and addressing their concerns was seen as vital.


*They were very good at consulting with the parents in what we wanted and what we needed and then they would have done their plans around that.*
(Mother T)

The social aspects were also valued for both the child and the family.


*They got a wee buddy system as well where he was going out with his friend, he met a wee friend a couple of days and the two of them went out together.*
(Mother T.)


*The most successful aspect of the project is seeing families come together and be able to participate in family activities which all family members can be included in. *
(Project staff)

The trusted relationship between the project and the social workers who had referred families brought benefits to both.


*There’s brilliant relationships with us and Cedar... there’s a two-way flow of communication. When a family is known to Cedar... that family does not need to contact us. They do not seem to need us. Their issues have been dealt with it. That really allows our social workers to deal with families with even more complex needs. It’s been a real resource to us in that way: to staff as well as to the families.*
(Trust staff 4)

#### 3.5.2. Improvements for the Project

Parents would have liked the project to have continued for longer and for more family days to be provided. Reduced waiting times for a place on the project was also noted. Project staff would welcome more contact with schools so that the strategies used in school and at home could be shared and closer links nurtured between schools and parents.

## 4. Discussion

This project is unique in a number of respects and the lessons from it can inform the provision of post-diagnostic support services for families whose children have ASD.

The project focused on promoting the social inclusion of children within their families and the local community. These two settings —the family and the community—provided the context in which children’s skills could be enhanced and practical support could be provided to families. Yet this approach stands in contrast with the focus on therapeutic approaches often used with children who have ASD [2]. Nevertheless, the focus on social inclusion brought about other specific gains to the child and to families as shown by the issues that were resolved during the project. Moreover, equipping children and families with the skills needed to function socially has potentially longer-term gains for the child as the Logic Model for the project identified (see Figure 1).

The project aimed to support families as well as the child especially ‘low-resource’ parents that the project had targeted. The most commonly expressed need was for information about available services and supports. This needs to be provided on an ongoing basis as parents’ needs will change over time. Regular home-based contact with parents created a trusted and more intimate relationship between staff and parents, which clinic visits or occasional parent-teacher meetings would find hard to replicate. Two outcomes are worth underlining. Parents’ self-confidence was boosted, which is a necessary prelude for them to instigate new ways of interacting with their child and trying new approaches. Additionally, many parents reported a reduction in stress within the family as they became more adept at managing the child’s behaviour and meltdowns. Hence, interventions solely focused on the child will not necessarily bring about the practical and emotional supports that parents need [7].

The choice of home-based supports was not just for practical reasons. Although, in rural areas, it overcame the lack of transport options available to low resourced families in particular. Rather, having project staff coming to the child’s home ensured that all the project work was personalised to the individual needs of the child and family [14]. Although children with ASD may share some common features, there were marked differences in how ASD was manifested in even this relatively homogenous sample of children. When the diversity found among families is added in, then the need for individualised interventions become ever more apparent. Admittedly, home visits are a more costly option than group-based parent training sessions. Yet, this is the only alternative presently offered to most parents in Northern Ireland and likely elsewhere. However, the uptake of group-based training is low especially for low resourced families and, to date, there is limited evidence as to the effectiveness of such training [15].

The project aimed to address the needs of siblings of the child with ASD who are often overlooked in ASD interventions. By contrast, parents are very aware of the impact the child with ASD has on their other children. Going to the family home meant that the staff could include the siblings in activities designed to help the child. The organisation of family days was another means of involving siblings in play and recreational activities. Both of the foregoing were arguably more successful that organising sessions for groups of unrelated siblings, which had been tried even though other studies have shown sibling support groups to be effective [16]. These may work better for teenage siblings whereas the siblings of the children in the project were mostly under 12 years of age. Nonetheless, the main message is that family interventions have to extend beyond parents to embrace siblings as well.

As with any innovative project, there are inevitably improvements that could be made to the service. Currently, the demand for it exceeds the places that can be available at any one time and, with increasing numbers of children being identified, this situation will worsen [17]. One option would be to reduce the length of time families are visited by the project or to increase the time between visits. Both options would allow more families to be accommodated for the same cost. Future research and evaluation could test out these options even though the solution is more likely for projects to become adept at adjusting their service to family needs and outcomes rather than having equivalent service inputs across all families.

The longer-term impact of the service also bears further study. There is evidence that early preschool intervention for children with developmental delays does result in longer term legacies [18], but this has yet to be determined for older children with ASD as well as for their families.

Evaluation methods could also be improved if the necessary resources were made available by the commissioners of new services. Pre-test and post-test measures of the children and parents would provide more robust evidence of change, as would the recruitment of a ‘waiting list’ control group to identify the improvements that might occur over a period of time even without any intervention.

## 5. Conclusions

In conclusion, post-diagnostic support for children with ASD and their families is vital. Providing cost-effective ways is a priority as is gathering evidence to show its impact. Staff trained in ASD but from non-clinical backgrounds, such as in this project, are an effective means of providing home-based community support to children and families.

## Figures and Tables

**Figure 1 brainsci-10-00318-f001:**
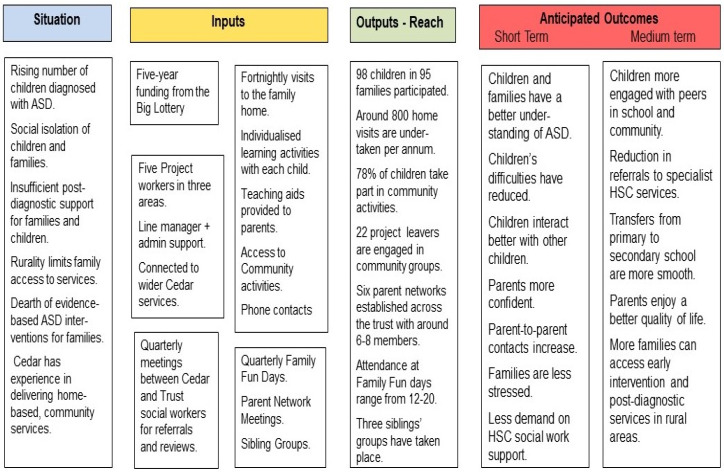
The logic model for the family-centred intervention.

**Table 1 brainsci-10-00318-t001:** The number and percentage of children rated by staff on outcomes achieved.

Child Difficulties	Never Had a Problem	Was a Problem—Getting Better since Start of Project	Still a Problem at End of Project
Difficulty in relating to other children and in making friends	0 (0%)	68 (81.0%)	16 (19.0%)
Awareness of dangers, road safety	8 (9.6%)	58 (69.9%)	17 (20.5%)
Difficulty with change	10 (12.0%)	56 (67.5%)	17 (20.5%)
Joining in community activities	12 (14.3%)	59 (70.2%)	13 (15.5%)
Anxious, agitated, nervous	18 (22.8%)	53 (64.6%)	10 (12.7%)
Extreme fear and nervousness, lack of confidence, depressed	33 (39.3%)	45 (53.6%)	6 (7.1%)
Anger, temper tantrums, meltdowns	34 (40.5%)	40 (47.65%)	10 (11.9%)
Problem with following instructions	36 (42.9%)	38 (45.2%)	10 (11.9%)
Personal care (toileting, dressing)	50 (59.5%)	23 (27.4%)	11 (13.1%)
Difficulties in communication: speech and/or language	51 (62.2%)	21 (25.6%)	10 (12.2%)
Issues with school, homework, etc.	53 (63.1%)	25 (29.8%)	6 (7.1%)
Problem with bedtime, sleeping	58 (69.0%)	15 (17.9%)	11 (13.1%)
Unusual interest in toys or objects	66 (78.6%)	10 (11.9%)	8 (9.5%)
Problems with play, keeping self-occupied	67 (79.8%)	14 (16.7%)	3 (3.6%)
Eating	69 (82.1%)	5 (6.0%)	10 (11.9%)
Unusual response to something new	69 (83.1%)	13 (15.7%)	1 (1.2%)

**Table 2 brainsci-10-00318-t002:** The number and percentage of families rated by staff on outcomes achieved.

Issues Families Can Face	Issues that were NOT a Concern	Project Helped and No Longer an Issue	Project Gave Some Help but Still an Issue
Knowing what services and supports are available to parents and children	3 (3.5%)	70 (82.4%)	11 (12.9%)
Managing the child’s behaviour, temper tantrums, and meltdowns	23 (27.7%)	38 (45.8%)	22 (26.5%)
Having time to spend with my other children	25 (29.1%)	50 (58.1%)	11 (12.8%)
Taking the child out of the house, joining in community activities	28 (32.6%)	40 (46.5%)	17 (19.8%)
Communicating with schools	36 (41.9%)	41 (47.7%)	9 (10.5%)
Relationships with siblings (or other children)	36 (42.9%)	32 (38.1%)	16 (19.0%)
Finding activities all the families can join in	37 (43.0%)	34 (39.5%)	15 (17.4%)
Worries about the child’s future	41 (47.7%)	7 (8.1%)	37 (43.0%)
Lack of confidence in how to manage my child	46 (47.9%)	20 (23.8%)	18 (21.4%)
Meeting other parents and sharing experiences	44 (51.2%)	38 (44.2%)	4 (4.7%)
Understanding what it means to have Autism/ASD	51 (59.3%)	29 (33.7%)	6 (7.0%)
Family quality of life	65 (79.3%)	12 (14.6%)	5 (6.1%)
Main caregiver often feels anxious or depressed	70 (81.4%)	3 (3.5%)	13 (15.1%)
Main caregiver has people to turn to if s/he has a problem	74 (86.0%)	7 (8.1%)	5 (5.8%)

**Table 3 brainsci-10-00318-t003:** The themes and subthemes reported as outcomes of the project by parents.

Main Themes	Subthemes	Supporting Quotes
Impact on the children	Social Interaction	N is an only child and his social skills were lacking. But whenever she would have taken him out and interacted him with other children as well, we could see a very big change in the social skills (Mother S).
	Improved behaviour	He was very frustrated, would have lashed out a lot, he would have cried and screamed a lot. So over time, she built up taking him out for like for a half an hour at one of the wee local centres … where anybody could come in with their children. And he actually started interacting with the kids. (Mother TR)
	Acquisition of new skills	The project helped my child to understand a lot of topics including personal safety, peer pressure, and safe strangers (Mother TA)
Impact on families	New learning for parents	The project was a massive help to my son and our whole family, to help us understand his condition and work together as a family to help him. (Mother DH)
	Increased confidence	They give us the confidence to think that you’re not doing a bad job … you’re doing your best. They were able to make people feel more confident in themselves that ‘I can do this’. (Mother H).
	Free time	I have a little six-year-old too. It’s very difficult for her when you have a little autistic child so, it gave me a bit of time with her. And she also went out too with the Cedar person at times, which also gave me a bit of time to do things about the house or go and do a bit of shopping and stuff like that. (Father).
	Meeting other parents	There was a family day and then a thing at Halloween and … you’re meeting other parents there as well with children who are similar, you know, so that’s quite good so it is. (Mother JO).

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
