# Peer review of "Promoting the Social Inclusion of Children with ASD: A Family-Centred Intervention"

_brainsci, 2020, doi:10.3390/brainsci10050318_

Round 1

Reviewer 1 Report

Dear authors!
The reviewed paper considers the acute and socially significant topic of the socialization of children with autism.
An undoubted strong side of the study is a comprehensive Description of Inputs and activities.
However, the methodology and characteristics of the sample in this study raised multiple questions.

1. What was the average household income? 
Line 156 - “Over two thirds had a WAGE EARNER in the family but over one quarter were dependent on social security benefits” ... Around half of the families (51%) lived in the top 30% of socially deprived areas in Northern with very few families coming from more affluent areas. Thus the project had recruited and retained under-resourced families which was its intention. ”
Here the sample which consists of 2/3 of the families which have domestic workers is in the top 30% of socially deprived and under-resourced families?

What was the number of children in these families? 
In line 154: “In all, 95 families with 98 children with ASD were involved with the project” Line 162 “Of the 96 children ...” - so how many children with ASD were examined in total?

How old were the children in your study? Line 11-12 declares “A project in Northern Ireland provided post-diagnostic support to nearly 100 families 12 and children aged from 4 to 11 years.” Line 163 “range 3.4 to 11.8 years”
Then it is more appropriate to say "from 3 to 11 years". And it is better to specify the exact age for each case.

How many project workers were considered? Line 96-98 says: “Three project workers: ONE fulltime AND TWO part-time job shares each covered one of three counties in the western part of Northern Ireland which is largely rural with a higher proportion of under-resourced families.” and thereafter in line 174-175: “The first author ... He conducted face-to-face interviews with all the project staff (7 in all)”
If 7 is the number of interviews, then please rephrase this sentence.
Line 174-178 - How many families were surveyed out of 98? 
Line 174 - What does “mix of qualitative and quantitative data” mean?
If it is a mixed-methods instrumentary, please clarify the methodology as follows.

What quantitative data include? What were the methods for analyzing it? What do qualitative data include? Which survey methodology was used? Which tools were used to process the results? Where are the questionaries?
At least a brief description should be given.
All the abovementioned issues should be resolved in detail in section 2.

Line 197 - How was this rating arranged? Please, clarify the methods here. 
Why it is not given in the Methods section?

A significant point of my criticism is the general negligence of the paper design. I see wrong alignments in width, extra spaces and their lack where they are needed. The quality of the figures and tables are to be improved.
Lines 48, 53, 61, 119, 125,152 etc. - different fonts in one line

Line 201 - “Insert Table 1 about here” Table I seems missing
Line 239 - “Insert Table 2 about here” Table II seems to be duplicate

Despite my overall positive impression, without a proper description of the methodology, the reported results are vague and the paper is not scientifically sound. So far, this is not a scientific article, but an essay describing the experience of talking with families with children with ASD. I recommend completely rewrite the paper in accordance with referees' recommendations.

Author Response

What was the average household income?  

This sensitive  information was not sought from families.  Rather it was obtained indirectly as described. 

Line 156 - “Over two thirds had a WAGE EARNER in the family but over one quarter were dependent on social security benefits” ... Around half of the families (51%) lived in the top 30% of socially deprived areas in Northern with very few families coming from more affluent areas. Thus the project had recruited and retained under-resourced families which was its intention. ”Here the sample which consists of 2/3 of the families which have domestic workers is in the top 30% of socially deprived and under-resourced families? 

Yes - the comparison is with socio-demographic make-up of the area in which families lived. 

What was the number of children in these families? 
In line 154: “In all, 95 families with 98 children with ASD were involved with the project” Line 162 “Of the 96 children ...” - so how many children with ASD were examined in total?

Apologies for the typos.  The correct numbers on which data was available was 92 families with 96 children. 

How old were the children in your study? Line 11-12 declares “A project in Northern Ireland provided post-diagnostic support to nearly 100 families 12 and children aged from 4 to 11 years.” Line 163 “range 3.4 to 11.8 years”
Then it is more appropriate to say "from 3 to 11 years".

This change has been made. 

And it is better to specify the exact age for each case. 

The age of children was based on their exact age when they started on the project.  

How many project workers were considered? Line 96-98 says: “Three project workers: ONE fulltime AND TWO part-time job shares each covered one of three counties in the western part of Northern Ireland which is largely rural with a higher proportion of under-resourced families.”

There were three fulltime positions but with five staff (two job shares).  This has been clarified. 

and thereafter in line 174-175: “The first author ... He conducted face-to-face interviews with all the project staff (7 in all)”
If 7 is the number of interviews, then please rephrase this sentence.   

Seven is the number of interviews - The five project workers and two managers. Line 176 clarifies this.

Line 174-178 - How many families were surveyed out of 98? 

See above, it was 92

Line 174 - What does “mix of qualitative and quantitative data” mean?
If it is a mixed-methods instrumentary, please clarify the methodology as follows.
What quantitative data include? What were the methods for analyzing it? What do qualitative data include? Which survey methodology was used? Which tools were used to process the results? Where are the questionaries?
At least a brief description should be given. All the above mentioned issues should be resolved in detail in section 2.

Lines 179-191 make clear how the qualitative and quantitative data was obtained and the other issues the reviewer noted. 

Line 197 - How was this rating arranged? Please, clarify the methods here. 
Why it is not given in the Methods section?

In response to similar comments from Reviewer 3 we have relabelled the ratings. To avoid unnecessary repetition we chose to give the details when reporting the results. 

A significant point of my criticism is the general negligence of the paper design. I see wrong alignments in width, extra spaces and their lack where they are needed. The quality of the figures and tables are to be improved.
Lines 48, 53, 61, 119, 125,152 etc. - different fonts in one line

We could not find these different fonts to which the reviewer refers in our copy of the paper.

Line 201 - “Insert Table 1 about here” Table I seems missing
Line 239 - “Insert Table 2 about here” Table II seems to be duplicate

These typos have been corrected.

Reviewer 2 Report

Thank you for the opportunity to read your very interesting paper. This is an under developed area of work so the project is a valuable contribution. I hope that you will find the following comments useful. For ease of following, I am working through the manuscript by line number.

Line 30  - Although the opening statement is quite factually correct, some connection to published work will add strength.

Line 34/35 - Over 60% - reference here as well please

Line 72 - After ASD - this may be an uploading or formatting issue but there are several spots where there are spaces.

Line 77 - Reference 9 - this is a quote requiring a page number.

Figure 1 - there are a couple of phrases that are likely quite common in your daily parlance but may not be as well known to a broader audience- e.g. WHSCT; leavers

Line 98 -I am assuming they all had bachelor level psychology - please be specific

Line 116 - I am using this paragraph as an example. There are places where present and past tense are used inconsistently. For example line 123 uses present tense whereas the paragraph above uses past tense. This is a minor issue but makes reading that much easier.

Line 151 - siblings - there was involvement from all 3 counties I presume but given the prior line it might be wise to specify.

Line 174 - How were the participants for interview chosen? Why this number? Recruitment protocol? 

Line 179 - could you offer a bit more in what is meant by pro formas.

Lines 185-188- Thank you for the ethical statement. I am not from Northern Ireland so this may be different. In my jurisdiction this same option exists for program internal evaluation but it does not permit external journal publication. Could you clarify that is permitted in your case as it will not be a necessarily common permission in some jurisdictions. As well, can you please describe the consent process which would include permission related to this publication.

Line 219 -should this not be labelled Table 1. It would also be helpful for Tables 1 and 2 to explain why the N differs in each line.I am sure the explanation is simple but it may not be obvious to some readers. Was there any statistical relevance/significance  analysis completed? If not (and that may well be appropriate) it would be useful to explain this. It may well be that frequencies are the best reporting method but why?

Line 270 - the method for thematic analysis has not be described. This should be here. It would be quite helpful to know.  In line with this, were interviews recorded and then transcribed? Or were hand notes used in which case are these direct quotes. 

Are all of the participants included in each thematic grouping? There are occasions where words like she or they are used. In parenthesis could the role of that person be noted e.g. line 276.  

the qualitative results are quote heavy. I wonder about editing / reducing so that the reader gets a more pointed sense of the theme as opposed to relying on the quote. This tends to leave us more wrapped up in the personal experience of the participant as opposed to getting a sense of the theme and why it matters.

Thank you for sharing this work.

Author Response

Line 30  - Although the opening statement is quite factually correct, some connection to published work will add strength.

A reference has been added.

Line 34/35 - Over 60% - reference here as well please

The reference given to the first part of the report has been moved to cover this comment.

Line 72 - After ASD - this may be an uploading or formatting issue but there are several spots where there are spaces.

This has been corrected.

Line 77 - Reference 9 - this is a quote requiring a page number.

The page number has been added.

Figure 1 - there are a couple of phrases that are likely quite common in your daily parlance but may not be as well known to a broader audience- e.g. WHSCT; leavers

These have been clarified in the Figure.

Line 98 -I am assuming they all had bachelor level psychology - please be specific.

The word 'bachelor' has been added. 

Line 116 - I am using this paragraph as an example. There are places where present and past tense are used inconsistently. For example line 123 uses present tense whereas the paragraph above uses past tense. This is a minor issue but makes reading that much easier.

The past tense is now used throughout this section.

Line 151 - siblings - there was involvement from all 3 counties I presume but given the prior line it might be wise to specify.

This has been clarified.

Line 174 - How were the participants for interview chosen? Why this number? Recruitment protocol? 

This detail has been added: 

telephone interviews were conducted with 10 parents chosen by project staff to represent the range of children and families involved in the project across the three counties.  Self-completion questionnaires were completed by a further 15 families who responded to an invitation sent to all parents by project staff as a text message.

Line 179 - could you offer a bit more in what is meant by pro formas.

The word pro formas has been changed to rating scales and reference added to Tables 1 and 2.

Lines 185-188- Thank you for the ethical statement. I am not from Northern Ireland so this may be different. In my jurisdiction this same option exists for program internal evaluation but it does not permit external journal publication. Could you clarify that is permitted in your case as it will not be a necessarily common permission in some jurisdictions. As well, can you please describe the consent process which would include permission related to this publication.

From the advice we received, there is no prohibition on publishing evaluation studies in journals.  However we have now made clear that consent was sought from parents. Lines 188-190 read:  "Families gave consent for the information they provided directly or indirectly to be used anonymously in any reports on the project internally and externally, for example to project funders".

Line 219 -should this not be labelled Table 1.

Apologies for the typo. It has been labelled Table 1.

It would also be helpful for Tables 1 and 2 to explain why the N differs in each line.I am sure the explanation is simple but it may not be obvious to some readers.

This explanation has been added in lines 204-205: "some items were omitted for some individuals due to uncertainty or irrelevance for the child or family, hence the totals do not add to 98".

Was there any statistical relevance/significance  analysis completed? If not (and that may well be appropriate) it would be useful to explain this. It may well be that frequencies are the best reporting method but why?

This was a descriptive report of issues affecting children and families and we had not undertaken any comparisons or relationships that required statistical testing.   It is now made clear that the evaluation was based on descriptive data.  (line 175).

Line 270 - the method for thematic analysis has not be described. This should be here. It would be quite helpful to know. In line with this, were interviews recorded and then transcribed? Or were hand notes used in which case are these direct quotes. 

The text now reads: This information was obtained mostly through one-to-one interviews which were audio-recorded and transcribed and supplemented by self-completed questionnaires.   A thematic content analysis using the framework proposed by Braun & Clarke [13] was undertaken.  

Are all of the participants included in each thematic grouping? There are occasions where words like she or they are used. In parenthesis could the role of that person be noted e.g. line 276.  

This comment is addressed by the new Table.- see below

the qualitative results are quote heavy. I wonder about editing / reducing so that the reader gets a more pointed sense of the theme as opposed to relying on the quote. This tends to leave us more wrapped up in the personal experience of the participant as opposed to getting a sense of the theme and why it matters.

The text has been replaced by a table as proposed by reviewer 3. 

Reviewer 3 Report

See attached file:

Round 2

Reviewer 1 Report

Dear authors,
Thank you for providing the revised manuscript and answers. I still think that the detailed and fundamental description of the chosen methodology could enrich your study. However, the formal criteria for publication are met now. I still can observe strange table placeholders in the paper, please, find some pictures enclosed.

Author Response

I have removed the place-holders that the reviewer mentioned.

Reviewer 2 Report

Thank you for the revisions - the article is much stronger - the addition of Table 3 is very good. The reduction of direct quotes from participants makes the ones used more impactful.

The article is ready from my perspective

Author Response

We appreciate your endorsement and thanks for your time and advice.